# Contrastive Learning for Image Registration in Visual Teach and Repeat Navigation

**DOI:** 10.3390/s22082975

**Published:** 2022-04-13

**Authors:** Zdeněk Rozsypálek, George Broughton, Pavel Linder, Tomáš Rouček, Jan Blaha, Leonard Mentzl, Keerthy Kusumam, Tomáš Krajník

**Affiliations:** 1Artificial Intelligence Center, Faculty of Electrical Engineering, Czech Technical University in Prague, 166 27 Prague 6, Czech Republic; brouggeo@fel.cvut.cz (G.B.); lindepav@fel.cvut.cz (P.L.); tomas.roucek@fel.cvut.cz (T.R.); jan.blaha@fel.cvut.cz (J.B.); leonard.mentzl@fel.cvut.cz (L.M.); krajnt1@fel.cvut.cz (T.K.); 2Department of Computer Science, University of Nottingham, Jubilee Campus, 7301 Wollaton Rd, Lenton, Nottingham NG8 1BB, UK; keerthy.kusumam2@nottingham.ac.uk

**Keywords:** visual teach and repeat navigation, long-term autonomy, machine learning, contrastive learning, image representations

## Abstract

Visual teach and repeat navigation (VT&R) is popular in robotics thanks to its simplicity and versatility. It enables mobile robots equipped with a camera to traverse learned paths without the need to create globally consistent metric maps. Although teach and repeat frameworks have been reported to be relatively robust to changing environments, they still struggle with day-to-night and seasonal changes. This paper aims to find the horizontal displacement between prerecorded and currently perceived images required to steer a robot towards the previously traversed path. We employ a fully convolutional neural network to obtain dense representations of the images that are robust to changes in the environment and variations in illumination. The proposed model achieves state-of-the-art performance on multiple datasets with seasonal and day/night variations. In addition, our experiments show that it is possible to use the model to generate additional training examples that can be used to further improve the original model’s robustness. We also conducted a real-world experiment on a mobile robot to demonstrate the suitability of our method for VT&R.

## 1. Introduction

For mobile robots, navigation is a fundamental competence. To navigate efficiently, it is necessary to perceive the surrounding environment and to know which direction to take in order to stay on the planned path. There are many approaches to perceiving the world and obtaining this type of information. Most autonomous mobile robots rely on sensors such as cameras or lidar. Lidar is popular due to its precise depth estimation and robustness to external lighting conditions. Due to their passive nature, camera-based systems typically do not achieve comparable robustness to illumination variations. On the other hand, cameras are cheaper, lighter, smaller, and more common. While lidar perceives the environmental structure in a relatively sparse way, camera data have higher resolution and provide information about a given scene’s texture and overall appearance. Due to these advantages, cameras are very popular as onboard sensors for mobile robots, and they are very popular for localization and navigation. To mitigate their dependence on lighting conditions, they are typically combined with other sensors, such as lidar, radar, GPS, odometry, etc. [1,2].

The challenges arising from unstructured and changing environments [3,4,5] are currently being addressed by many researchers in the fields of robot localization, mapping, and navigation. Appearance changes are particularly challenging for cameras because they can occur abruptly, while the environmental structure itself changes slowly (e.g., shadows or weather variations). The dynamics of appearance variations makes visual navigation in natural environments a challenging task. Moreover, long-term localization in slowly changing environments [6,7] is also difficult by itself. Commonly, even the most robust landmarks or features can disappear or change their appearance so much that they cannot be associated with their previous counterparts [8]. Especially tricky is capturing seasonal changes in nature due to foliage variations and appearance changes between day and night in urban areas. In these cases, the whole scene can have different coloring: trees appear very different due to present or absent foliage, likewise when illumination changes from abundant natural diffuse light to sparse point lights from street lamps and vehicles.

Significant efforts have been taken to craft stable image features by hand [9,10]. After the boom of deep learning, it became apparent that the method that extracts stable image features can be learned rather than handcrafted [11]. To achieve proper scene understanding, it is desirable to produce representations of images that represent the location’s semantics rather than appearance. In the past, contrastive learning proved to be particularly efficient for learning such representations [12]. In this work, we aim to use these learned representations for visual teach and repeat (VT&R) navigation, where a robot has to move autonomously along a path it had previously been guided along.

Contrastive learning methods are often referred to as self-supervised or weakly supervised. The significant improvement over the classic supervised setup is that the network can create representations to distinguish between individual scenes, and it is not bound to any categories created by annotators. Moreover, large-scale datasets for these methods do not require extensive labeling. Weak supervision in these methods is achieved by providing the network information about the similarity or dissimilarity of specific pairs of images. In recent years, several works have presented various types of self-supervised pipelines focused on creating robust image representations using CNNs or attention-based neural networks [13,14].

In this paper, we propose using Siamese neural networks for the purpose of VT&R. Using four datasets containing drastic changes in environmental appearance, we demonstrate that their robustness to appearance variations outperforms state-of-the-art methods tailored for the purpose of VT&R navigation [15]. Moreover, we propose a novel, self-supervised training pipeline for the Siamese networks that allows carrying out the training process with automatically generated annotations.

The following section discusses the current state of research in VT&R, contrastive learning, and machine vision in robotics. Then, in Section 3, the proposed method is described, and Section 4 following defines the evaluation methodology. Further, in Section 5, the comparison of our model to other methods is presented, and real-world experiments demonstrate the validity of our method. Finally, in Section 6, we discuss the method’s results and outline further possible improvements.

## 2. Related Work

Teach and Repeat (T&R) is a popular framework that facilitates the robot to autonomously navigate a previously traversed path. In the teaching phase, the robot is shown the desired path, and during the repeat phase it attempts to closely follow that path. Even though the method may not be as general as classic SLAM, since the choices for paths have to be pre-defined, it achieves significantly higher robustness and therefore applicability [16]. Most popular approaches for teach and repeat navigation utilize landmarks and/or keypoint descriptors that are either handcrafted [17] or learned [15,18] in order to precisely estimate the heading correction [17] for the robot by computing the differences in the camera views and the images captured during the teach and repeat phases.

Several previous works have shown that heading displacement can be estimated from visual data alone in order to traverse a recalled path [19,20]; this was further theoretically proven via the convergence theorem [21,22]. This can bypass the need to apply advanced methods to retrieve the exact geometric transformations between the current location and the locations available in the data acquired during the teaching phase. Vision-based methods are useful at this point to compute the horizontal displacement of the current view relative to the images collected during the mapping phase. Such systems do not need to be informed of the source of the displacement, which allows them to be readily applicable in scenarios where the robot encounters lateral or heading displacements.

DallÓsto et al. demonstrate in their recent work that horizontal displacements estimated from the cross-correlation between the camera image and the map image can be employed for a robot’s path traversal [23]. However the approach offers limited robustness in changing environments since it does not directly use any learned transformation of the images to accommodate those changes. Several approaches exist to tackle teach and repeat navigation in changing environments. In most scenarios, the robot uses a learning algorithm that captures the changes in the environment. Such lifelong [24] learning is capable of adapting robot behavior according to gathered experience [25]. One of the possible solutions is to perform several traversals of the taught path under varying conditions, such as weather or time of day, and compare the current camera view to that of the images collected from all other traversals. Other approaches include modeling the environmental dynamics and modifying the behavior accordingly [26] or predicting the changes [8,27].

Most of the currently used approaches use feature-based vision methods. A known drawback of feature-based approaches is that they may fail to provide enough correspondences that are repeatable and stable across longer time periods [8,26]. In addition, when we want to produce a large number of features, time and memory complexity of the computation grows rapidly. Feature-based methods usually do not provide complex information about probability of all possible displacements out of the box. Previously described methods also lack any prior knowledge about the world and rely on multiple drive-throughs to create robust representations of particular places.

In this paper, we examine another popular approach in robotics, where we first learn good representations from sensor data on large-scale datasets and then deploy the trained models for robot perception [28]. We also include a transfer learning step where the models are fine-tuned during the run to specialize them towards the deployed environment. The Siamese network design is suitable for this workflow, as it can capture similarities and variations in the scene over long time periods [29]. CNNs are adept in learning features that are robust to the environment and viewpoint changes, according to [30,31,32,33]. Simultaneously, it is feasible to learn to select features that contribute towards accurate heading estimation and reject those which are not significant.

There is remarkable progress in the field of deep neural networks designed for solving computer vision problems. State-of-the-art visual representations are learned using transformer architectures such as ViT [34] or DETR [35] that are inspired by the success of attention mechanisms popular in natural language processing. Research also shows how to efficiently train widely used CNN architectures such as EfficientNet to match the performance of transformer models [36] by optimizing design decisions such as filter sizes, skip connections, normalizations, and regularizations using augmentations.

Siamese networks were initially developed to learn differences in data, especially when the data are multi-modal or come from different sources such as RGB-optical flow, depth maps, audio, etc. They are not bound to a particular domain and can be used to process faces, time-series, or signatures [37,38]. A pair of data inputs are fed to the backbone network, which processes each of the data points in two forward passes, therefore sharing its weights. The output features of the Siamese backbone are then compared depending on the task, and the network updates its parameters based on the direction estimated by a contrastive loss function. The contrastive loss aims to learn which data points are similar to each other. This simple framework delivers powerful representations that achieve state-of-the-art performance in visual recognition tasks [13,39]. Another advantage of this method is that it can be deployed as a one-shot learner where only one example is necessary to recognize similar samples representing the same class [40,41]. Contrastive learning can also be used to obtain dense representations that are robust to daily and seasonal changes [42,43,44].

We hypothesize that a fully convolutional Siamese network can effectively estimate the difference in heading between two images. These networks are widely used for object tracking [45,46]. They can generate high fidelity information about object displacement with high efficiency, so it is possible to achieve real-time performance. This makes them a suitable choice for the task of image registration, which is crucial for VT&R navigation [15,21,22]. However, training of Siamese networks normally requires high-quality annotations, which are expensive and laborious to acquire. We address this issue by creating a semi-supervised training pipeline that is able to annotate the training data.

## 3. Method Description

For the purpose of horizontal registration of the images, we use a fully convolutional Siamese network. It is possible to use various CNN architectures as the backbone. We use a relatively shallow network with only five layers. The size of the backbone is kept lightweight so that the network can be easily deployed on robots with limited computing capability. The backbone is designed to reduce the width of the input eight times. An input image of 512 pixels is reduced to a representation 64 bins wide. This level of compression is a good compromise between accuracy and robustness. The situation is different for the height of the image. We squeeze the height as much as possible, making the network more robust to vertical shift of the image pair.

There are no fully connected layers at the end of the backbone so that the representation generated by the backbone stays position-dependent. This is crucial for our method, as otherwise final cross-correlation between them cannot be implemented. This immediately reduces the number of possible backbone architectures, and our experiments have shown that our training pipeline is not suitable for usage of attention mechanism. It is undesirable to find distant relationships prior to the cross-correlation of the representation, and it causes very low loss on training data without any meaningful generalization for the evaluation set.

First, the pair of images is passed through the backbone. The backbone outputs neural representations of the images. One of these representation is then considered as a kernel which is applied to the second representation by cross-correlation. As one representation is shifted along the other, the method outputs a cross-correlation for all possible shifts. The shift with highest cross-correlation is then used as the displacement estimate. Output of whole network for images *A* and *B* can be written as:(1)$$f\left(A,B\right)=f_{b}\left(A\right)\;*\;f_{b}\left(B\right),$$
where $f_{b}$ is a function that is trained by the backbone network, and its output gives us the neural representations, which are then convolved (∗) together.

### 3.1. Training

We describe the training process as follows. First, a pair of images taken from a similar place is chosen. One of the images from the pair is then cropped. We opt to crop to 56 pixels wide, so its neural representation (backbone output) has a width of 7 bins. The network is then trained by estimating the position of the crop in the second image. This process is shown in Figure 1.

The loss function *L* to train the network is the binary cross-entropy:(2)$$\begin{array}{cc} l_{n} & =y_{n}\log x_{n}+\left(1-y_{n}\right)\log\left(1-x_{n}\right) \end{array}$$(3)$$\begin{array}{cc} L & =\frac{\sum_{n=0}^{N}l_{n}}{N}, \end{array}$$
where $x_{n}$ is the *n*th element of the network’s output vector bounded by the sigmoid function between [0,1], and $y_{n}$ is the binary target, which must have the same size *N* as the cross-correlation of the representations and reflects where the cutout is located on the second picture.

The proposed training scheme has many advantageous properties. Firstly, our pipeline is able to craft representation suitable for registering the images, which is our main target in this paper. As a byproduct, these representations can also be used to align images with higher precision. Secondly, it is unnecessary to deal with network collapse because the parts of the images not present in the crop can be considered negative samples. Finally, the pipeline is relatively simple and easy to implement. It also does not have excessive computation demands, and one GPU with a large VRAM is sufficient for the training.

If we want to achieve as high accuracy and robustness as possible, it is vital to create a target that addresses both properties. We conducted experiments with three different methods of target creation. The first option we tried was a narrow target with a single positive value in the center of the crop. The second option was a wide target with ones inside all the bins where the crop was located and zeros everywhere else. The first option showed a lack of robustness, while the second caused problems with accuracy. To address this, we created a spiky target, as shown in Figure 2. The spiky target has the maximum in the middle of the crop and then decreases linearly towards the crop’s edges.

### 3.2. Inference

The displacement prediction pipeline is shown in Figure 3. The main difference between the inference and training phase is in the size of the images. We do not use any cropping for the inference—whole images are used. Convolution is then done between neural representations of whole images. When whole images are used, the final cross-correlation can exploit the long-range relationships between the neural representations.

When the network’s inputs are the whole images, adding roll (cyclic) padding to both sides of one of the image representations is necessary. The padding is needed to enable the network to output a certain range of displacements. If a different type of padding is used (zeros or ones), the method tends to produce biased outputs towards the center or towards the edges. Another possibility is to crop one of the images even during the evaluation, but this means that the final cross-correlation cannot exploit the long-range dependencies, and it caused inferior performance in our preliminary experiments. An example of network prediction is shown in Figure 4.

### 3.3. Semi-Supervision

To train our system on how the environment can change, we need to collect images of the same scene at different times. A popular dataset for this purpose is the Nordland dataset [47], which consists of four videos taken from a train traveling from the south of Norway to the north. Each of these four videos is taken in a different season and captures the changes in nature over the year. Another advantage is that the data contain the GPS position of the train so that it is possible for us to find the pairs of images taken from a similar place. Further, the train goes on the rails, so there is minimal heading displacement between corresponding images. However, the Nordland dataset also has many issues. First of all, the rails are very distinctive, and if they are kept in the training samples, the network can easily be fixated on the rails. Further, the Nordland dataset lacks the day/night differences and has only brief footage of urban areas.

Alignment accuracy between training images is crucial to achieving a smooth learning curve and finding proper kernels inside the backbone. GPS accuracy is not sufficient enough to train the network correctly. For this reason, we first train the network using only coarsely aligned images by way of GPS data. This model is then used to obtain training pairs with better alignment. In Figure 5, we show how crops are chosen depending on the precision of the dataset.

In Figure 6, it can be seen that the model can be used to improve image alignment compared to solely using GPS positions. In both videos, the indices of images with similar GPS locations are sampled. An array of tuples of corresponding indices $\left(i_{k},j_{k}\right)$ is obtained. The variables *i* and *j* are indices of corresponding images in the *k*th sampled tuple. In one video, we directly use images with index $i_{k}$. In the second video, we use images from the interval $\left[j_{k}-D,j_{k}+D\right]$ concatenated into one batch. The constant *D* defines the search span. GPS drift should not be higher than *D* in terms of the number of images. The Siamese network then compares the image from the first video to the whole batch of images from the second video. The network outputs the likelihoods for each image in the batch from the second video. These likelihoods are used as a similarity measure between images, and the image with the highest likelihood is chosen as the best match.

Although the Nordland dataset is useful, it has many issues that have already been discussed in this section. We used the same procedure for the UTBM Robotcar [48] dataset to address these issues. This dataset contains drives through the city over one year. Unlike Nordland, the UTBM dataset contains footage from an urban area with many vehicles and day/night changes. Switching from train to car poses an additional issue with the vehicle’s heading. We are using the already trained network to predict the heading displacement between the drives. So the model is effectively self annotating the training data. Looking at the likelihoods of predicted heading, we can evaluate the annotation quality and drop invalid pairs. Despite this, alignment accuracy is still much better than in the Nordland data.

### 3.4. Application in VT&R

We find the architecture suitable for application in robotics for several reasons. First of all, the exact transformation between the images is not needed because our network is trained end-to-end to register the images horizontally. This approach significantly simplifies the whole computing pipeline, which makes it easy to train. Another benefit of our method is that we obtain the likelihood of all displacements out-of-the-box. This is especially advantageous for probabilistic localization methods such as a particle filter [49], where probabilities of multiple states of the robot can be considered. Moreover, it is very common that even incorrectly classified images have a significant peak of likelihood at the correct place. Further, our model is very efficient. We achieved such performance with the minimal size of the network so that it can be used in real-time. We offloaded all the computation to the external GPU (NVIDIA GTX1050), and we measured performance around 25 FPS while retaining very low memory demand. This computation speed is sufficient for real-time performance on a real robot.

There is also no need to save all the images during the teaching phase. In the descriptor setup, we usually save only the keypoints. Similarly, for our method, it is feasible to save only the output of the backbone. This neural representation can be up to eight times smaller than the original image. As we show in the next chapter, the representations can also be used to find the most-relevant image in the sequence of images from the recorded map, i.e., the image captured at the closest location.

### 3.5. Limitations of the Method

Even though we assume a locally planar drive surface, the method must be robust to some roll and vertical-shift changes between images. Currently, the network can handle only limited roll differences. To make the network robust to these transformations, we can apply them artificially to the data during neural network training. Additionally, cyclic padding can also be added in the vertical dimension of the image representation, which can further improve robustness to vertical shift between images. It is easy to do the cross-correlation over the whole batch, so it is possible to create artificially rolled images of the map and process them with a small overhead in inference time.

The method is also not suitable for image retrieval or place recognition. On the one hand, unlike methods using visual words to compare the images, our approach is sensitive to small changes in the view that help distinguish spatially close images. On the other hand, the neural network is not trained to look for unique visual words to help recognize certain places. Thus, our method is better suited to search for the best-matching place in shorter sequences.

Furthermore, semi-supervision can be used to improve the robustness of the network to specific conditions or environments. It was possible to get very accurate image pairs from the Nordland dataset because we did not need to estimate the train’s heading. On the contrary, when data are collected from a wheeled robot or car, the displacement must be estimated together with the closest image pair. These annotations can be only as accurate as the model, so they cannot improve model accuracy.

Finally, the method is intended for deployment in VT&R systems based on visual servoing principles. Therefore, our method is not suitable for full 3D or 6DoF position estimation.

## 4. Experimental Methodology

In this section, we will go through the methodology used in the experiments. Firstly, we discuss the backbone’s architecture and the hyperparameters used to train the Siamese network. Then, we summarize the datasets used in the paper. Further, the description of methods used for comparison is given. Finally, we describe how the methods are evaluated and qualitatively compared.

### 4.1. Network Architecture and Training

The whole neural network is implemented in PyTorch [50]. We used Adam optimizer [51] for stochastic gradient descent. We trained the neural network with multiple hyperparameter settings. The method has multiple hyperparameters, and the best results we achieved for the values of hyperparameters are written in Table 1. We used Bayesian hyperparameter optimization to find the optimal values. The system with current architecture must be trained with relatively high learning rates, and the sweep also showed that very large batch sizes are beneficial for training stability. The higher number of channels at the end of the backbone can improve performance, but it poses additional memory demands on the device storing the representations, which is undesirable.

We also trained the system with various backbone architectures. Adding standard residual blocks with skip connections [52] does not improve the performance of the model. This is probably caused by the small size of the model. Squeeze and excitation layers [53] and layer pooling can both improve performance, but only by a small margin. The final version of the architecture is shown in Figure 7. It is impossible for us to reuse commonly used small neural networks (VGG-16, ResNet-18) because of dimension-reduction constraints. While there is the possibility of reusing a few initial layers from these pre-trained models, our initial experiments showed that this leads to poor performance since the kernels are typically trained for image classification, which is a different task. There are additional hyperparameters to the training scheme, such as number of hard negative samples or width of the target. Hard negative samples consist of image pairs which do not correspond and the target has only zeros. We use 1/4 as a fraction of hard negative samples in every training batch. The width of spike in the target shown in Figure 2 should correspond to the width of the cropped image.

### 4.2. Training and Testing Datasets

All the training was carried out on the rectified Nordland and UTBM Robotcar datasets. For the evaluation, we used four additional challenging datasets. Three of them—Stromovka, Carlevaris (subset of the NCLT dataset), and Planetarium—have already been presented in [15,54,55]. The fourth dataset is from a parking lot in the Prague district of Čestlice and contains temporal changes, occlusions, and day/night differences. The summary of datasets used is provided in Table 2. Each of these datasets has images captured in a different season from similar positions and with a slightly different heading. All the image pairs have been annotated with pixel-wise displacements.

All the evaluation datasets have been annotated by humans. Each pair of images has corresponding keypoints labeled as shown in Figure 8. A pair of images might not be taken from exactly the same place. This can cause ambiguity in the annotation. We take the average horizontal displacement of annotated correspondences to evaluate the network. This also puts a lower bound on the precision of the dataset. For example, on Stromovka there is, on average, a 17-pixel standard deviation between individual annotations in a pair of images. The Planetarium dataset is much better suited to evaluate accuracy—the images are taken from very similar positions and have less than two pixels standard deviation of annotations. However, Planetarium contains images with distinctive human-made objects, which reduces the dataset’s difficulty.

### 4.3. Methods to Compare

We compare our approach with the one based on the GRIEF [15] descriptor. GRIEF was trained for seasonal invariance and has performance comparable to the Superpixel method [56]. As a baseline, we also used the SIFT descriptor [9] as well as the Superpoint [57]. However, the released Superpoint model is trained on the MS-COCO dataset only, and it is not tailored to seasonal changes. Thus Superpoint can be tested only on the Carlevaris dataset. Keypoint-based algorithms use a histogram voting scheme to calculate the horizontal displacement of the image pairs. In particular, they extract image features from the currently perceived image and the one from the map and establish their correspondences as described in [9] or [15]. Then, they calculate the differences in the horizontal coordinates of each feature pair. Finally, they calculate the modus of these differences, which is then considered an estimate of the horizontal displacement *d* of the given image pair.

Siamese networks do not break the images into local features, but use cross-correlation of the outputted representations of both images. Since the input images of our network are re-scaled to 512 pixels and its output is 64 pixels wide, we use cubic interpolation to re-scale the output back to match the width of the processed images. The predicted horizontal displacement *d* is then calculated from the position of the maximum of the recalled output.

### 4.4. Performance Evaluation Methodology

The aim of the performance evaluation was to compare how accurately the individual methods determine the horizontal shifts *d* between the image pairs of the testing datasets. To compare the performance of individual methods on a particular dataset, we followed the methodology used in the paper [58]. This paper used a qualitative and a statistical way to compare the accuracy of image registrations. First, we calculated horizontal displacements ${}_{i}d_{s}^{m}$ of each image pair *i* of the dataset *s* and the method *m*. This displacement was then used to establish registration errors ${}_{i}e_{s}^{m}$ as absolute values of the difference between the registration provided by a method *m* and ground truth of each image pair, i.e., ${}_{i}e_{s}^{m}={|}_{i}d_{s}^{m}{-}_{i}d_{s}^{gt}|$, where ${}_{i}d_{s}^{gt}$ is displacement of the image pair *i* established by manual annotation (gt stands for `ground truth’). Thus, the performance of the method *m* on dataset *s* was characterized by a sequence describing its registration errors $e_{s}^{m}$.

The resulting sequences $e_{s}^{m}$ were then processed to compare them statistically and qualitatively. To determine which of the methods performs statistically significantly better, we applied a paired Wilcoxon test over the sequences belonging to the same dataset. In other words, to determine if a method *a* performed differently than method *b* on dataset *s*, we used a paired Wilcoxon test to reject the hypothesis (at significance level of 3%) that the sequences $e_{s}^{a}$ and $e_{s}^{b}$ were similar. If the methods were found to perform differently, we calculated their mean average error (MAE) to determine which of the methods performed better.

To compare the methods qualitatively, we estimated cumulative distribution functions of the registration error sequences $e_{s}^{m}$ for each method and dataset. These functions indicated if the probability that the registration error of a method *m* on dataset *s* was lower than a given threshold *t*, i.e., $F_{s}^{m}\left(t\right)=P\left(e_{s}^{m}\leq t\right)$. Displaying the $F_{s}^{m}\left(t\right)$ of multiple methods in the same graph provided more intuitive insight into their performance than the rigorous statistical tests. Moveover, following the methodology of [15,59], we used the value of $F_{s}^{m}\left(32\right)$ as a ratio of accurate predictions (RAP). Similar to [59], the choice of t=32 pixels corresponded to the width of two bins in the outputted representation.

## 5. Experimental Results

Using the methodology described in Section 4, we performed a comparison of the proposed method to other state-of-the-art algorithms. Moreover, we investigated the influence of training set selection and performed a field experiment where the method was used to guide a mobile robot along a previously taught trajectory in an environment undergoing a drastic appearance change.

### 5.1. Performance Comparison to Other Methods

Our first investigation compared the performance of our method to the ones based on SIFT [9], GRIEF [15] and Superpoint [57]. Thus, we calculated the sequences $e_{s}^{m}$ for each method and processed them as specified in Section 4.4. The cumulative distribution functions $F_{s}^{m}\left(t\right)$ are shown in Figure 9. These figures indicate that the Siamese network attains superior performance compared to the other tested methods on all testing datasets. The Siamese network has better precision and robustness. The Wilcoxon test also confirmed that the Siamese networks achieve statistically significantly lower errors than the other methods at the significance level of 3%.

### 5.2. Impact of Training Dataset Selection

We also investigated how the choice of the training dataset impacts the performance of the Siamese network and summarized the results in Table 3. The neural network itself rectified both training datasets, and, since it was much easier to align images taken from the train, the Nordland dataset had much better precision of training pairs. This can be seen especially in the Carlevaris and Planetarium datasets, where adding the UTBM dataset into training resulted in worse precision. It can also be seen that even coarsely aligned datasets created by the neural network can improve performance. Such a new dataset could be useful, mainly when it contains a novel type of data. The most significant improvement was when the network was trained on the compound dataset and evaluated on the Čestlice dataset. This is to be expected because Čestlice and UTBM are both distinctly urban with both day and night changes. The UTBM dataset also improved robustness on different types of data, such as Stromovka and Carlevaris.

One of the significant advantages of our method is that it outputs similar representations for scenes with different appearances. Thanks to this good encoding, we achieved state-of-the-art performance for horizontal image registration on all the evaluation datasets. Our method is designed explicitly to predict the horizontal displacement, so is not as versatile as keypoint matching, but it can provide estimates with high accuracy and robustness.

### 5.3. Vision-in-the-Loop Experiment

To verify the feasibility of the method in a real robotic system, we integrated it into a prototype navigation system of a car transporter at the Škoda car manufacturing plant in Mladá Boleslav, Figure 10.

The vehicle is equipped with a camera, but its navigation is primarily based on its laser rangefinders and differential GPS. To evaluate our VT&R method, we performed two experiments. In both of the experiments, the vehicle was guided by its camera only, and its position was measured by a Leica TS16 Total Station. The robot, which has an Ackermann drive, used the calculated displacement obtained from the neural network to directly control the angle of its front wheels.

During the first experiment, we used images captured during the day of a 100 m long path segment that the vehicle traverses during its routine operation. Then, we have disabled the vehicle’s GPS and lasers and displaced it from the recorded path by ∼2 m. After that, we let it traverse the path at night using our VT&R method. The path was recorded during the day, but the robot was traversing at night, which resulted in drastic differences in appearance, Figure 11. Despite this, the robot reduced its initial displacement during the 100 m traversal as indicated in Figure 12.

Furthermore, we performed another evaluation similar to the setup in [21,22]. We taught the robot a 100 m closed path during the day. Then, we displaced it by ∼4 m, let it traverse the path autonomously, and measured its displacement from the taught path with a Leica TS16; we recorded a value every time it completed a loop. Figure 13 shows that the robot gradually decreased the displacement with each traversal both at night and day, eventually stabilizing the displacement below 0.2 m.

In both of the experiments, we also tested the SIFT- and GRIEF-based image registration methods, but they could not guide the robot reliably at night using the map taught during the day. These experiments demonstrated the generalization ability of the presented method as it was trained on the Nordland dataset aimed at seasonal changes rather than day/night variations.

## 6. Conclusions

We presented a novel approach for image registration between images captured at the same location under significantly varying conditions. Our method is aimed for usage on autonomous mobile robots, in particular for visual teach-and-repeat navigation. The evaluation of our method indicates that it outperforms state-of-the-art algorithms on multiple datasets with seasonal changes and day/night appearance variations. We also demonstrated that our method can be used for other downstream tasks such as searching for the most similar image in a sequence of images, which can be used in teach-and-repeat schemes such as [60]. This ability also allows the creation of alignment-rectified datasets, which can be used for further training of the neural network. Moreover, the presented model exhibits desirable properties for VT&R navigation, such as providing the likelihoods of all possible displacements out-of-the-box or the ability to process images at a high framerate with low computational demands. To facilitate the use of our method and allow its reproducibility, we have disclosed the source codes at https://github.com/Zdeeno/Siamese-network-image-alignment (accessed on 11 April 2022).

Our research suggests that it is possible to see which parts of the image contributed to the displacement prediction. Suppose the operation between the image representations is element-wise multiplication with average over the channels instead of matrix multiplication, the parts with the highest contribution to the current prediction yield the highest values. In this way, it is possible to create heatmaps that can be further analyzed and used for advanced scene processing. Another possible improvement could be achieved by employing 360° cameras. Such an image can be divided, and the displacements would be predicted over the image’s parts. This could give us valuable insight into the current displacement of the robot from the intended trajectory and bring us closer to 6DoF localization along the path.

## Figures and Tables

**Figure 1 sensors-22-02975-f001:**
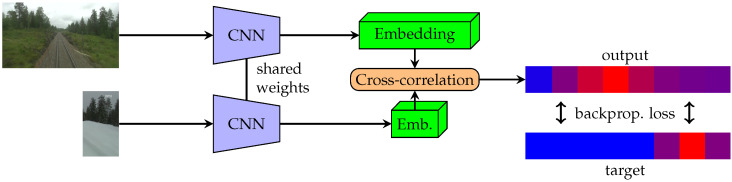
Diagram depicting the scheme of the training pipeline for the Siamese network models presented in the paper. Training is done over pairs of image and a cutout of another image of the same scene at a different time. The network itself then consists of the backbone with shared weights and a final cross-correlation layer which gives the final output.

**Figure 2 sensors-22-02975-f002:**
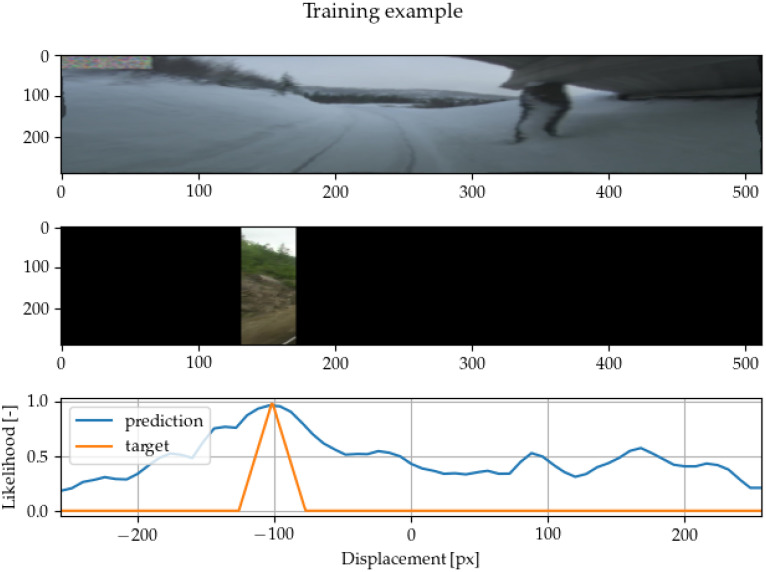
An example of the training process of the alignment method. The sample was generated using one image and a random cutout of the second image. Plotting the network output against the training target, one can see that the network is already approximating it reasonably well.

**Figure 3 sensors-22-02975-f003:**
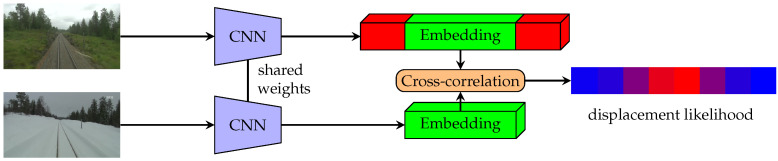
Diagram depicting the prediction pipeline for the methods presented. The inputs are two standard images, whose embeddings are then convolved together to get the final prediction of displacement. Note the additional roll padding in the representations.

**Figure 4 sensors-22-02975-f004:**
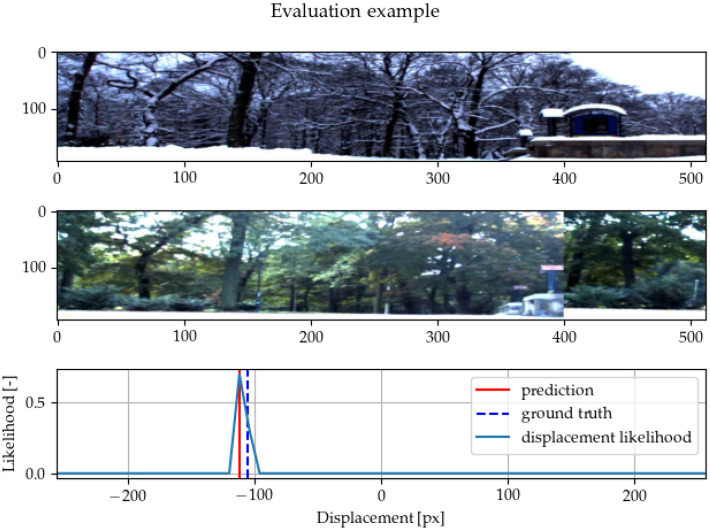
An example of the method correcting the horizontal alignment on an image pair from the Stromovka dataset used in [15]. Two images of the same scene are given to the network for alignment. There is visible shift (predicted by the network) with the roll padding in the bottom image. The likelihood of different potential displacement as outputt by the network is shown in the last diagram. In this example, the images are well-aligned.

**Figure 5 sensors-22-02975-f005:**
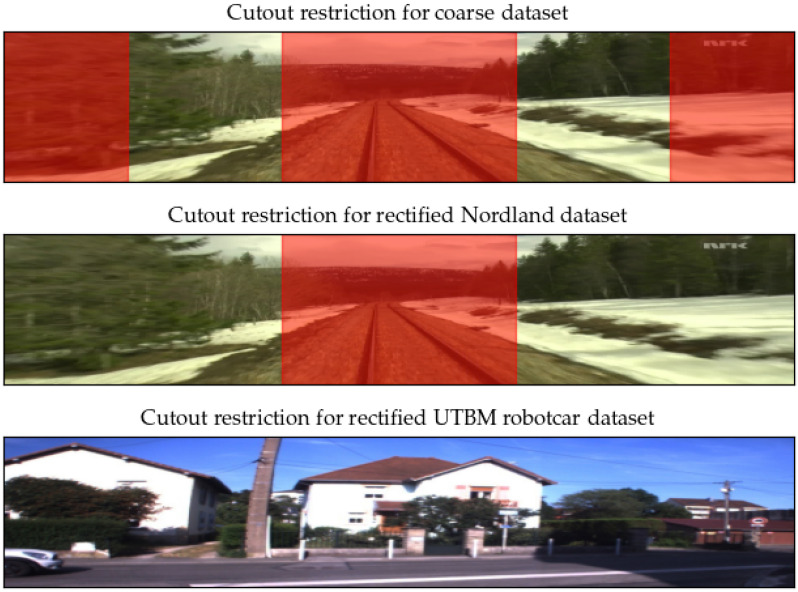
For individual datasets, different restrictions of candidate cutout regions have been used. The coarse dataset is designed to focus on distant objects, in the rectified dataset, we only limit the network from learning to align using the rails, and the EU long-term dataset is used as is.

**Figure 6 sensors-22-02975-f006:**
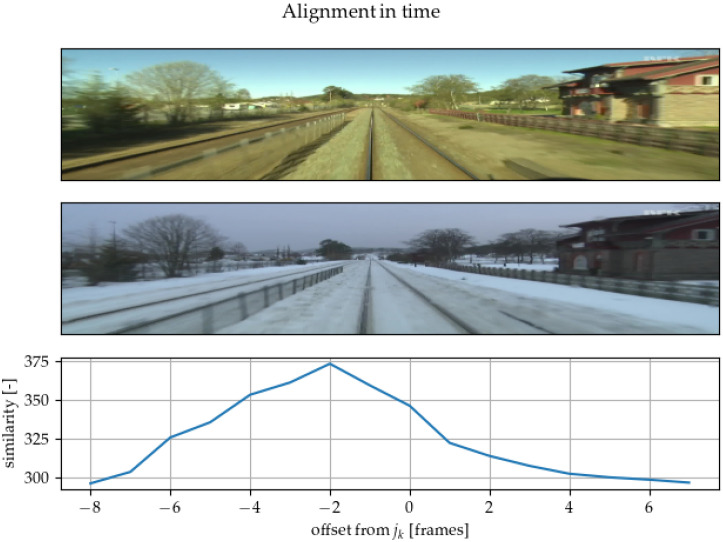
Using a similarity measure, the position of individual images from GPS annotations have been corrected along the path. In this pair, GPS-based positioning is quite precise, so the correction is only minus two frames.

**Figure 7 sensors-22-02975-f007:**
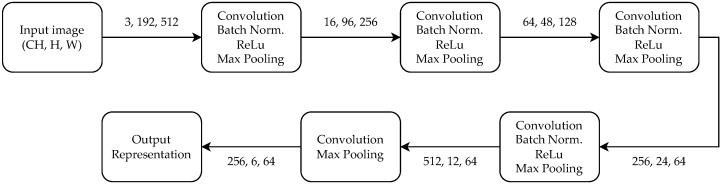
Visualization of backbone architecture and the dimensions of the tensors. It can be seen that at the end of the network is pooling used only for height to reduce the sensitivity to vertical shift between images.

**Figure 8 sensors-22-02975-f008:**
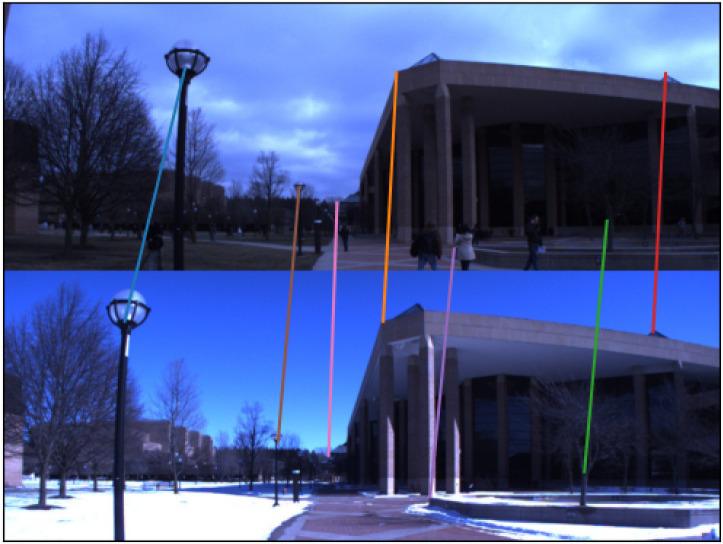
A sample of annotated image pair from Carlevaris dataset. The annotations are visualized as color lines that connect the two images’ correspondences. Both images are taken from a similar place, so the individual annotations have a slight variance in displacement correspondence in the *x*-axis.

**Figure 9 sensors-22-02975-f009:**
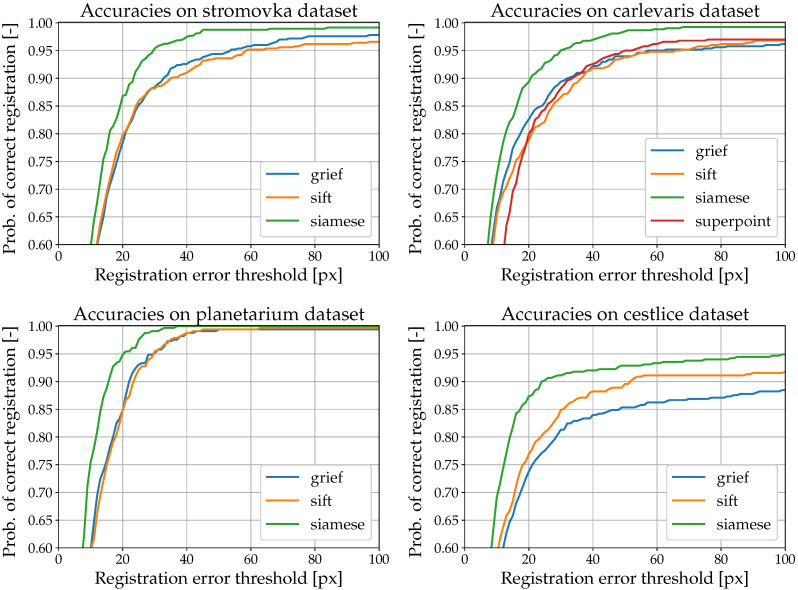
Cumulative distribution functions of the registration error achieved for the individual methods and datasets. Cumulative distribution was calculated as the ratio of samples where the method estimated the displacement in pixels to be under a certain threshold (shown on the *x*-axis). A detailed description of the cumulative distribution function calculation is provided in Section 4.4. For every dataset, Siamese networks outperformed other methods.

**Figure 10 sensors-22-02975-f010:**
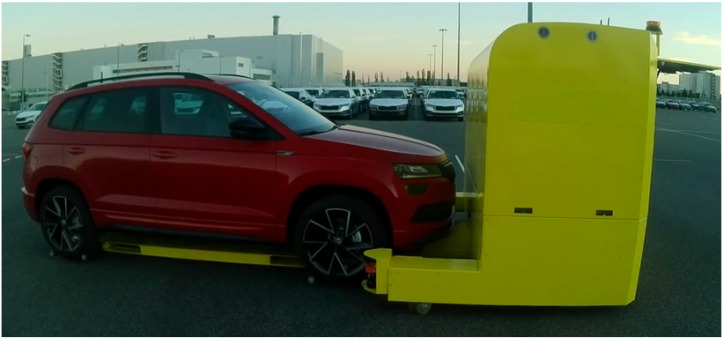
Autonomous vehicle prototype intended for car and heavy load transportation used for the vision-in-the-loop experiments. Courtesy of [44].

**Figure 11 sensors-22-02975-f011:**
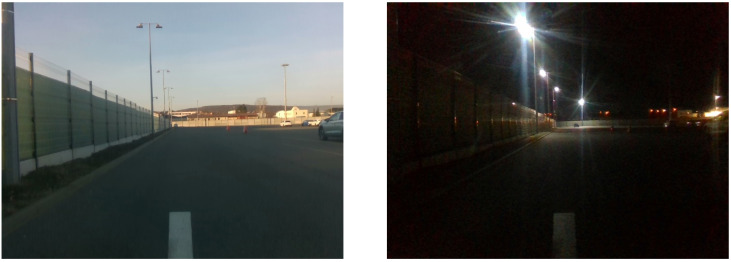
Example of an alignment task of two images with significant time and conditions difference captured at the location of the experiment. The daytime image was used for teaching; the nightime image was part of the repeat phase deployment.

**Figure 12 sensors-22-02975-f012:**
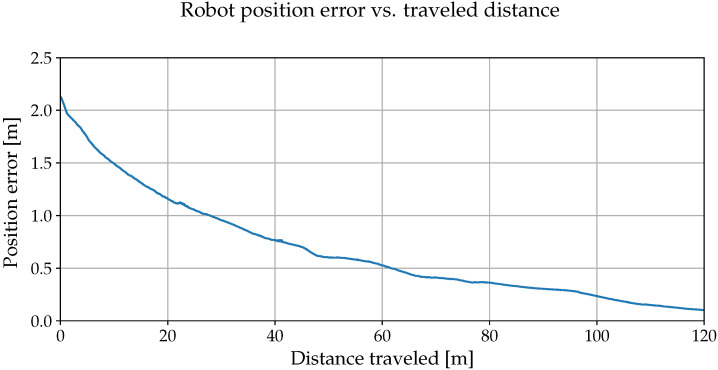
The lateral position error along a 100 m long path and its evolution as our method autonomously steered to correct it.

**Figure 13 sensors-22-02975-f013:**
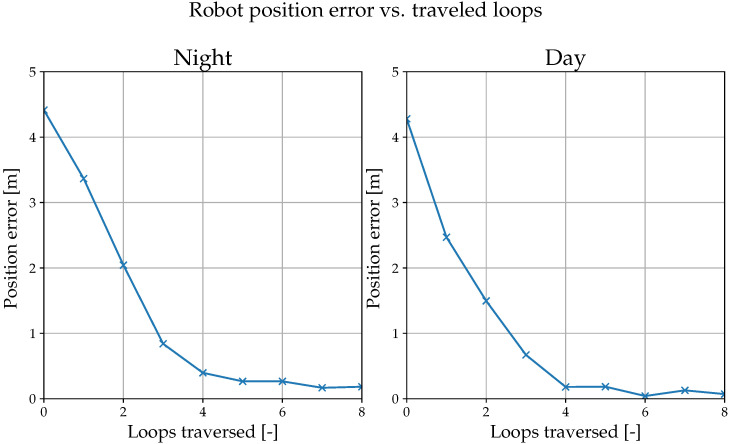
The lateral position error while traversing a closed path. Marked points indicate that the vehicle traveled the full loop and was starting a new traversal.

**Table 1 sensors-22-02975-t001:** Hyperparameters used for training of the best model.

Hyperparameter	Value	Units
Learning rate	10${}^{-4}$	[-]
Batch size	96	[-]
Cutout size	56	[px]
Epochs	150	[-]
Number of layers	5	[-]
Filter size	5	[px]
Output channels	256	[-]

**Table 2 sensors-22-02975-t002:** This table summarizes all the information about datasets used. Each dataset contains images taken from the same place across multiple seasons. All the possible unique combinations of seasons are used.

Dataset Name	Type Changes	Seasons / Places Num. of Pairs	Usage Avg Annotation Std.
Nordland [47]	nature seasonal	4/14,000 84,000	training self-annotated
UTBM EU [48]	urban day/night	10/500 22,500	training self-annotated
Stromovka [15]	nature seasonal	2/500 500	evaluation 17 pixels
Carlevaris [54]	nature+urban seasonal	2/500 500	evaluation 15 pixels
Planetarium [15]	nature+urban seasonal	12/5 330	evaluation 2 pixels
Čestlice	urban day/night	10/10 450	evaluation 2 pixels

**Table 3 sensors-22-02975-t003:** Results when trained on different datasets, with both mean average error (MAE) in pixels and ratio of accurate predictions (RAP) calculated as described in Section 4.4. The threshold for ratio of accurate prediction (RAP) was set to 32 pixels, which corresponds to two bins in the outputted representation. Best results are indicated in bold.

	**Training Set**							
	**Nordland Only**			**UTBM Only**		**Nordland + UTBM**		
**Evaluation Set**	**MAE**	**RAP**		**MAE**	**RAP**		**MAE**	**RAP**
Stromovka	12.64	95.4%		29.02	85.2%		**12.27**	**96.2%**
Carlevaris	**18.67**	93.1%		26.17	85.5%		18.83	**94.2%**
Planetarium	**6.85**	**99.7%**		15.29	96.0%		6.99	99.4%
Čestlice	28.91	87.7%		54.04	73.5%		**20.49**	**91.6%**

## Data Availability

Datasets and code for the work presented here are available at this link: https://github.com/Zdeeno/Siamese-network-image-alignment (accessed on 11 April 2022).

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
