# Peer review of "Contrastive Learning for Image Registration in Visual Teach and Repeat Navigation"

_sensors, 2022, doi:10.3390/s22082975_

Round 1
Reviewer 1 Report
Visual teach and repeat navigation is popular in robotics thanks to its simplicity and versatility. Although the teach and repeat frameworks were reported to be relatively robust to changing environments, they still struggle with day to night and seasonal changes. This paper aims to find the horizontal displacement between recorded and currently perceived images, which is required to steer the robot towards the previously traversed path. Finally, a real-world experiment on a mobile robot was given to demonstrate the suitability of method for the VT&R.
On the whole, although the study of this manuscript is meaningful, there are still some problems in the arrangement of articles.
- The arrangement of figures and tables in the article is problematic. It is very chaotic and inconvenient for readers to see directly. Please revise according to the format requirements of the journal.
Generally speaking, it is often to describe "as shown in figure x or table x" in the text, and then give the figure x or table x as follows.
- The format of the table is incorrect.
- The data curve obtained in the article does not give a detailed explanation, which can not be summarized by the readers themselves.
- In the vision-in-the-loop experiment, based on Figure 10, it can be found that the brightness at night is relatively clear, and the comparison with the daytime experiment is not convincing. It is suggested to do the experiment in the environment with poor brightness at night to effectively verify the effectiveness of the method proposed in this paper.
- In the aspect of Data Availability Statement, the link is given, but I can not open the images, video and codes.
- The font size and picture size in some drawings are recommended to be revised to meet good visual effects.
Author Response
Dear reviewer,
the response is in the pdf together with edited document and highlighted changes.
Thank you, Zdeněk Rozsypálek

Reviewer 2 Report
This paper is about Contrastive Learning for Image Registration in Visual Teach and Repeat Navigation.
I have the following comments:
- Misspelling in the title (Naviagtion) should be (Navigation)
- Abstract “Visual teach and repeat navigation (VT&R) is popular in robotics thanks to its simplicity and versatility” -->there should be a comma or period before ‘thanks’
- The first reference in the Introduction starts with [2], where is reference [1]
- The Introduction section needs some modifications. What are the contributions of the paper? Also, at the end of Section 1, we usually see a section such as “The remainder of the paper is structured as follows: Section 2 ………”
- Before the statement “In this paper, we examine another popular approach in robotics ……” please write a section to state the limitations of related work.
- What are the parameters of the proposed CNN model?
- Why didn’t the authors use transfer learning models such as VGG-16, ResNet50, Inceptionv3, EfficientNet, etc.
- In the model description, Fig2, I do not see the authors used maxpooling layers. Please clarify
- The authors did not explain much about how hyperparameters are chosen. This is important as this may affect the performance. We usually use optimization techniques such as GA or grid search to optimize the hyperparameters. So, if optimization was not used, what was the method chosen to select hyperparameters?
- Write a section to describe the limitations of the work
- Write a section to describe the future work
Author Response

(The authors gave the same response as above.)

Round 2
Reviewer 1 Report
The author has made some modifications to the paper, but the modifications are not perfect, or many problems have not been modified. I think researchers should have a rigorous scientific research attitude.
On the whole, although the study of this manuscript is meaningful, there are still some problems in the arrangement of articles.
- The arrangement of figures and tables in the article is problematic. It is very chaotic and inconvenient for readers to see directly. Please revise according to the format requirements of the journal.
Generally speaking, it is often to describe "as shown in figure x or table x" in the text, and then give the figure x or table x as follows.
- The format of the table is incorrect, it should be in the form of three line grid.
- The data curve obtained in the article does not give a detailed explanation, which can not be summarized by the readers themselves.
- The values in some tables do not give units, for example, Figure 5.
- The font size and picture size in some drawings are recommended to be revised to meet good visual effects.
- Reference 1 should appear before reference 2, not from reference 2.
- In Section 5, the title 5.1 is given. What about title 5.2? If there is no the title 5.2, the title 5.1 should be deleted. Similar problems appear in the conclusion.
Author Response

(The authors gave the same response as above.)
